# Clinical Diagnosis of Rhabdomyolysis without Myoglobinuria or Electromyographic Abnormalities in a Dog

**DOI:** 10.3390/ani13111747

**Published:** 2023-05-25

**Authors:** Koen Maurits Santifort, Marta Plonek, Paul J. J. Mandigers

**Affiliations:** 1IVC Evidensia Small Animal Referral Hospital Arnhem, 6825 MB Arnhem, The Netherlands; 2IVC Evidensia Small Animal Referral Hospital Hart van Brabant, 5144 AM Waalwijk, The Netherlands; 3Department of Clinical Sciences, Faculty of Veterinary Medicine, Utrecht University, 3584 CM Utrecht, The Netherlands

**Keywords:** rhabdomyolysis, electromyography, myoglobinuria, tetraparesis, weakness, dog

## Abstract

**Simple Summary:**

In this case report, we describe the case of a dog with a clinical diagnosis of rhabdomyolysis without myoglobinuria or EMG abnormalities. Rhabdomyolysis is a potentially life-threatening condition characterized by the breakdown of skeletal muscle fibers and the release of their contents into the bloodstream. This can lead to a range of symptoms, including weakness, muscle pain, dark urine (myoglobinuria), and, in severe cases, kidney failure. In humans and dogs, diagnosis typically involves, among other things, the measurement of creatine kinase (CK) activity, an enzyme released during muscle breakdown, and urinalysis for myoglobin, as well as imaging studies and other tests, such as electromyography (EMG). The severity of rhabdomyolysis can vary to a large degree. Prompt diagnosis and treatment are critical for a positive outcome. While myoglobinuria is a common finding in humans and dogs with rhabdomyolysis, it may not always be present. Similarly, while EMG studies can be a useful tool for detecting muscle abnormalities and assessing the extent of muscle damage, results thereof may not always be abnormal in cases of rhabdomyolysis. Therefore, the absence of EMG abnormalities and/or myoglobinuria does not necessarily rule out rhabdomyolysis in dogs. This case report underlines the importance of not ruling out rhabdomyolysis as a cause of clinical symptoms in dogs without myoglobinuria or EMG abnormalities.

**Abstract:**

A 2-year-old female neutered Old German Shepherd was presented for acute non-ambulatory tetraparesis. Upon presentation to the emergency department, hematology and biochemical blood tests revealed no abnormalities aside from mildly elevated C-reactive protein levels (22.5 mg/L, reference range 0.0–10.0) and immeasurable creatine kinase (CK) activity. Neurological evaluation the next day revealed ambulatory tetraparesis, general proprioceptive deficits, mild ataxia and dubious diffuse myalgia. Withdrawal reflexes were weak on both thoracic and pelvic limbs. The CK was determined to be significantly elevated at that point (32.856 U/L, ref. range 10.0–200.0). Urinalysis revealed no abnormalities. An electromyographic (EMG) study of thoracic limb, paraspinal and pelvic limb muscles revealed no abnormalities. A magnetic resonance imaging (MRI) study of the cervicothoracic spinal cord was performed and revealed no abnormalities. A presumptive clinical diagnosis of rhabdomyolysis without myoglobinuria or EMG abnormalities was formed. Muscular biopsies were declined due to the rapid clinical improvement of the dog. A follow-up showed the progressive decline of CK activity to normal values and clinical remission of signs. A diagnosis of rhabdomyolysis was concluded based on clinical signs, consistent CK activity elevations and the response to supportive treatment for rhabdomyolysis, despite the absence of myoglobinuria and EMG abnormalities. Rhabdomyolysis should not be excluded based on the lack of EMG abnormalities or myoglobinuria in dogs.

## 1. Introduction

Rhabdomyolysis (RM) is a clinical syndrome of acute muscle fiber necrosis, classically characterized by swollen painful muscles (myalgia), weakness/collapse and myoglobinuria [1,2,3,4,5,6,7,8]. RM is a potentially life-threatening condition, primarily since it can induce acute kidney injury (AKI) [1,2,5,6,9].

There is some controversy in the literature on the exact definition of RM, with some authors including the above-mentioned clinical triad into the definition for the diagnosis of RM, while others focus on creatine kinase (CK) activity cut-off levels and the acute nature of the disorder [2,6]. In the conclusion of a systematic review of the human literature [6], a diagnostic definition for mild RM was suggested by the authors as “a clinical syndrome of acute muscle weakness, myalgia, and swelling combined with a CK activity cut-off value of >1000 IU/L or CK activity >5 times the upper normal limit”. To be classified as severe RM, the authors suggested the inclusion of the presence of myoglobinuria and a diagnosis of AKI. The authors also provided a ‘best descriptive approximation’: “*RM is a syndrome of acute or subacute onset in which a patient develops localized or generalized muscle pain and weakness, associated with a rapid increase in the serum CK activity, the extent of which will depend upon the timing of analysis with respect to the acute event*”.

Diagnosis typically involves blood tests to measure levels of muscle proteins such as CK, urinalysis to identify myoglobinuria, imaging studies such as magnetic resonance imaging (MRI) and electromyography (EMG), and muscle biopsy (although muscle biopsy is not regarded as obligatory for the diagnosis of RM in humans) [1,2,3,4,5,6,8,10,11]. Urinalysis is performed to assess for the presence of myoglobinuria, which is a useful and common clinical sign in cases of RM [1,2,3,4,5,6,7]. EMG may be used to assess for muscle damage (e.g., extent and distribution) in a less invasive manner than a muscle biopsy [10]. The possibility of a lack of EMG abnormalities and a lack of myoglobinuria is noted in the human literature [1,3,10,11], but has not been specifically documented in canine cases of RM. Indeed, comparatively little is known about RM in dogs compared to horses or humans [5]. In this case report, we describe the case of a dog with a clinical diagnosis of rhabdomyolysis without myoglobinuria or EMG abnormalities.

## 2. History and Case Presentation

A 2-year-old female neutered Old German Shepherd was presented for acute non-ambulatory tetraparesis. The referring veterinarian had administered a single oral dose of firocoxib 5 mg/kg orally 6 h before due to suspicions of the dog experiencing ‘diffuse/non-localizable pain’. The owner reported no possibilities for the ingestion of toxins, no witnessed traumatic events and had not administered any medication or antiparasitic treatment during the last week. The dog had gone for walks on a leash in the days leading up to and in the morning on the day of presentation. Upon presentation to the emergency department in the evening, a general physical examination revealed no abnormalities, though the dog was panting. A heart rate of 88/minute and a rectal temperature of 38.6 degrees Celsius were recorded. Hematology and biochemical blood tests (Table 1) revealed no significant abnormalities aside from immeasurable CK activity and mildly elevated C-reactive protein levels (22.5 mg/L, reference range 0.0–10.0). The dog was presented to the neurology department the next day, already having shown signs of improved motor function. Neurological examination revealed a bright, alert and responsive dog with ambulatory tetraparesis, general proprioceptive deficits and mild ataxia. The proprioceptive deficits were most obvious when testing the pelvic limb hopping responses and proprioceptive placement. Withdrawal reflexes were weak on both the thoracic and pelvic limbs. No overt signs of myalgia or swelling were noticed during the palpation of the paraspinal or limb muscles, although the dog was apparently anxious during the palpations. All other spinal reflexes were deemed to be normal. Cranial nerve function testing revealed no abnormalities. A diffuse neuromuscular disorder was suspected and an additional cervical spinal cord lesion was considered (although less likely) based on the proprioceptive deficits and the mild ataxia in conjunction with tetraparesis. Repeat testing of CK activity revealed a significantly elevated level of activity (32.856 IU/L, ref. range 10.0–200.0) (Table 1). Urinalysis (cystocentesis under sedation) revealed no abnormalities. An electromyographic (EMG) study (under general anesthesia premedication with medetomidine administered intravenously at 2 µg/kg and butorphanol at 0.3 mg/kg, followed by anesthesia with propofol administered intravenously at 10 mg/kg) of thoracic limb (triceps, biceps, extensor carpi radialis and palmar interosseous), paraspinal (thoracic and lumbar epaxial) and pelvic limb (quadriceps, biceps femoris, semimembranosus, tibialis cranialis and plantar interosseous) muscles on both sides revealed no abnormalities. In order to evaluate for the presence or absence of a cervical spinal cord lesion, which could possibly account for some of the clinical findings as mentioned above, and to assess for muscular changes in that area, an MRI study of the cervicothoracic spinal cord and surrounding structures, including muscles (C1–T6), was performed (under general anesthesia), including T2-weighted (T2W) sagittal plane, T1W sagittal plane, short-tau inversion recovery (STIR) sagittal plane, STIR dorsal plane, and T2* (gradient echo) and T2W transverse plane over the region of the C3–4, C4–5, C5–6, C6–7, C7–T1 and T1–2 intervertebral disc spaces. Evaluation of the images revealed no abnormalities of the spinal cord or surrounding structures, including musculature. A presumptive clinical diagnosis of rhabdomyolysis without myoglobinuria or EMG abnormalities was formed. Muscle biopsies were declined as the dog was already showing significant clinical improvement. The dog was treated with intravenous fluid therapy (lactated Ringer’s solution 4 mL/kg/h and sodium chloride 0.9% 5 mL/kg/h tapered during hospitalization over 24 h). Repeated blood tests showed the progressive decline of CK activity during hospitalization. Urinalysis remained unremarkable. The dog was discharged, having significantly improved neurologically and showing no signs of myalgia (either spontaneous or on palpation), and the owners scheduled follow-up examinations with the referring veterinarian. The CK activity had returned to within reference range 6 days after presentation to the emergency department (Table 1). The dog had recovered completely within that period without any residual neurological deficits. Over a 15-month follow-up period, no recurrences of clinical signs were reported, and the dog lived an unrestrained life including regular long walks (>60 min) and exercise (e.g., playing fetch).

## 3. Discussion

RM is most frequently diagnosed in humans after (over-)exertion, such as after performing some type of endurance sport [2,3,8,12]. In these cases, it is referred to as exertional RM. In veterinary medicine, RM is infrequent, but exertional RM in racing (grey)hounds or sled dogs is an exception [4,5,13,14,15]. In the veterinary literature, exertional RM is classified as a non-inflammatory, acquired myopathy [4,5]. Other etiologies for RM reported in the human and veterinary literature include intoxication, drug-induced, metabolic derangements, inherited metabolic disorders, physical causes (e.g., trauma, seizure-induced (also regarded as exertional) and hyperthermia), infection and with or without a genetic predisposition [1,2,3,5,16,17,18,19,20,21,22,23,24]. If no etiology is identified, the term ‘idiopathic’ is often applied. In the canine case we reported here, no clear etiology was identified. If RM episodes recur repeatedly after exertion, the term recurrent exertional RM is applicable (which has been extensively studied in horses) [1,2,3,4,5,6,7,8,25]. In the case reported here, no recurrences were noted over the 15-month follow-up period. Taking into account the CK activity cut-off of >1000 IU/L (or more than five times the upper normal limit) and the lack of signs of AKI or myoglobinuria, the definition of ‘mild RM’ was fulfilled [6].

Regarding the diagnosis of RM, serum CK activity is the most sensitive marker for muscle necrosis, and serum CK activity has been identified as a marker for the risk of developing AKI [1,2,5,8,26,27]. Its importance is underlined by the inclusion of cut-off values into clinical diagnostic definitions of RM in the human literature [6]. However, elevated CK activity levels are not specific to a diagnosis of RM [3,8]. Case history, clinical signs and the results of other diagnostic modalities must all be taken into account.

In humans with RM, EMG findings are often normal (up to 67% of cases, in which 74% of examined muscles showed normal EMG findings) [9]. When EMGs show abnormalities, they may be subtle, in contrast to other myopathies such as myositis [9]. The findings from EMGs in dogs with RM have not been reported in large studies. Abnormalities including fibrillation potentials, positive sharp waves and complex repetitive discharges are mentioned sporadically in case reports [28]. There are no large studies of dogs that report EMG findings. In the case reported here, the EMG of several muscles revealed no abnormalities.

While myoglobinuria is a common finding in cases of RM in both dogs and humans, it may not always be present [1,2,3,4,5,6,7,8,15]. Even if urinalysis does not reveal the presence of myoglobinuria, it is an important diagnostic tool since RM can lead to AKI, for which urinalysis can provide important information. Myoglobin is reported to be detectable in urine if serum levels exceed 0.3 mg/L or 0.5–1.5 mg/dL in serum [2,3]. There are several methods for the detection of myoglobin in urine, and the use of a urine dipstick is a quick, bedside option [2,5]. A positive result may be found when either hemoglobin or myoglobin are present in urine. Macroscopic examination (e.g., color) and the presence or absence of red blood cells (microscopically) can help to distinguish hemoglobinuria and myoglobinuria [2,5]. There are no large studies in dogs that report the incidence of myoglobinuria in cases of RM. In the case reported here, no myoglobinuria was identified.

Diagnostic imaging, particularly MRI, is important and is often used in human medicine to diagnose and prognosticate RM (as well as other myopathies), and to rule out other potential causes of muscle weakness and breakdown [11,29,30,31,32]. MRIs are useful for identifying areas of muscle inflammation, edema and hemorrhage. MRI findings in relation to RM most often include diffuse, patchy or multifocal hyperintensity on T2W and STIR images of affected muscles, as well as results indicative of hemorrhage (e.g., T1W hyperintensity). However, in some cases of RM, MRI studies may not reveal any abnormalities. In the case reported here, the MRI was primarily performed to exclude a (concurrent) myelopathy, since the ataxia and proprioceptive deficits in combination with tetraparesis were deemed suggestive thereof. No abnormalities were found in the cervicothoracic spinal cord, nor were any muscular abnormalities detected in surroundings muscles included in the field-of-view. In dogs, MRI has been used to diagnose various myopathies, but there are no large studies on RM in dogs that report MRI findings [28,33,34,35,36].

The clinical signs of proprioceptive deficits and ataxia in the dog reported here were curious. Proprioceptive deficits and/or ataxia have not been necessarily reported as clinical signs in cases of RM, though anecdotally they have been mentioned, mostly in cases when intoxication was the cause of the RM and likely also of the ataxia [20,22,24]. Several hypotheses could account for the (apparent) presence of these clinical deficits in the case reported here, and in RM in general. It may be that the release of muscle proteins and the subsequent inflammatory response that occurs in RM leads to nerve and (intrafusal) muscle fiber damage. This damage could then result in the loss of proprioception and the development of ataxia. Peripheral neuropathy has been reported as a sequela in humans with RM [30]. It could also be that the central nervous system function is somehow affected by the metabolic derangements that occur due to the RM. An alternative plausible explanation would be that the clinical signs were a misinterpretation, and that the apparent ataxia, delayed hopping responses and placing tests were due to muscle weakness. Finally, although not suggested by the clinical history or any diagnostic findings, intoxication cannot be totally excluded as no specific tests were performed to assess for toxins, and liver or renal function tests were not performed.

Regarding the treatment of RM, intravenous fluid therapy remains the cornerstone [2,5]. This is also true for (the prevention and treatment of) AKI [37]. Regarding the prevention of AKI, recent human medical guidelines recommend against the use of diuretics (such as furosemide or mannitol). In the case reported here, treatment was limited to intravenous fluid therapy. The dog had received a single oral dose of firocoxib, but as no clear signs of myalgia were noticed during hospitalization, this was not continued, nor was it replaced with other analgesic medications. The lack of clear signs of myalgia in this case could have been due to a number of reasons, including subsidence of most of the myalgia at the time of neurological investigation, coupled with the residual effects of the single dose of firocoxib, or the nature of the dog (i.e., propensity not to show signs of pain). The lack of clear signs of muscle swelling at the time of the neurological examination might be similarly explained. As the literature notes and clinical experience teaches, it can be difficult to assess some of the clinical signs of myopathies in dogs [38,39].

## 4. Conclusions

We reported a case of rhabdomyolysis in a dog without electromyographic abnormalities or myoglobinuria. Rhabdomyolysis should not be excluded based on the lack of electromyographic abnormalities or myoglobinuria in dogs. Although rhabdomyolysis in dogs can be lethal, mainly due to a chance of developing acute kidney injury, recovery may be swift and complete with supportive therapy (mainly fluid therapy) and the follow-up monitoring of markers of muscle damage and kidney function.

## Figures and Tables

**Table 1 animals-13-01747-t001:** Results of blood tests and urinalysis.

Day 1 (0 h) (Emergency Presentation)		Day 2 (14 h) (Presentation Neurology Department)	Day 2 (24 h)	Day 7
CBC ^1^ and differentiation	n.a.	-	-	n.a.
Biochemistry ^2^	n.a.	-	-	n.a.
CRP ^3^	22.5 mg/L	-	-	-
CK ^4^	- ^5^	32.856 IU/L	10.230 IU/L	103 IU/L
Urinalysis		Macroscopy: pale yellow, clear Dipstick: n.a., pH 6 Sediment: few epithelial cells	Macroscopy: pale yellow, clear Dipstick: n.a., pH 6	

^1^ CBC: complete blood count; ^2^ Biochemistry included glucose, creatinine, urea, total protein, albumin, globulins, alanine transferase, alkaline phosphatase, gamma-glutamyl transferase, total bilirubin, cholesterol, amylase, lipase and electrolytes (total calcium, phosphate, potassium, chloride, sodium); ^3^ CRP: c-reactive protein (ref. range 0.0–10.0 mg/L); ^4^ CK: creatine kinase; ^5^ CK on day 1 was immeasurable (no result obtained), n.a. = no abnormalities.

## Data Availability

All of the data are available in the present manuscript.

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
