# Peer review of "Clinical Diagnosis of Rhabdomyolysis without Myoglobinuria or Electromyographic Abnormalities in a Dog"

_animals, 2023, doi:10.3390/ani13111747_

Round 1

Reviewer 1 Report

Reviewer comments for manuscript ID animals -2335190 ‘Clinical Diagnosis of Rhabdomyolysis Without Myoglobinuria or Electromyographic Abnormalities in a Dog’

General Comments

Rhabdomyolysis is rarely encountered in canine clinical practice compared to horses and in canines too restricted to grey hounds involved in racing. Myoglobinuria and acute kidney injury are the frequent clinical pathological alterations are the common findings. Identification of specific biomarkers is the cornerstone of contemporary clinical pathology of many diseases and affections in veterinary practice. Absence of common clinical pathological alterations poses a challenge for diagnosis and treatment by clinicians. A rare clinical case has been presented by the authors where classical pathology of rhabdomyolysis was missing except the elevated C-Reaction protein and CK values. A thorough investigation based on the symptoms and clinical history has been done by the authors. The manuscript has been nicely presented and writing is almost flawless. The only limitation is just a single case report on which wider conclusions cannot be drawn. However, further work on this aspect of clinical pathology of rhabdomyolysis might follow this work.

Specific Comments

Lines 27 & 30: Please check the values for CK and C-phase protein. In one case the elevation is described mild and in the other significant. Please clarify.

Lines 95-100: No renal function and hepatic function tests were conducted to rule out any toxicosis. Please clarify.

Author Response

Dear Reviewer,

Thank you for your time and feedback. We have addresses your specific comments below and, where necessary, implemented the changes into the new version of the manuscript.

Kind regards,

Authors

Reviewer 2 Report

Dear Authors,

the manuscript is so well and understandably written. The authors nicely present a case of rhabdmyolysis in a dog, something more commonly seen in horses. Since there is a scarcity of cases of canine rhabdomyolysis this case report will undoubtedly make a contribution to clinical approach to this disease in dogs. 

the case report has been written according to the guidelines of the journal Animals. There is just one minor correction - in line 99, where the dose of medetomidine is reported the units should be changed from ug/kg to μg/kg, as to avoid possible confusion.

Author Response

Dear Reviewer,

Thank you for your time and feedback. We have addresses your specific comment and implemented the change into the new version of the manuscript.

Kind regards,
Author